# Thermoregulatory, Cardiovascular and Perceptual Responses of Spectators of a Simulated Football Match in Hot and Humid Environmental Conditions

**DOI:** 10.3390/sports11040078

**Published:** 2023-03-31

**Authors:** Johannus Q. de Korte, Thijs M. H. Eijsvogels, Maria T. E. Hopman, Coen C. W. G. Bongers

**Affiliations:** 1Department of Medical Biosciences, Radboud University Medical Center, 6500VC Nijmegen, The Netherlands; 2School of Sport and Exercise, Health Promotion & Performance, HAN University of Applied Sciences, 6525AJ Nijmegen, The Netherlands

**Keywords:** body temperature regulation, sports, extreme heat, heat stress disorders, hydration status

## Abstract

Major sporting events are often scheduled in thermally challenging environments. The heat stress may impact athletes but also spectators. We examined the thermal, cardiovascular, and perceptual responses of spectators watching a football match in a simulated hot and humid environment. A total of 48 participants (43 ± 9 years; *n* = 27 participants <50 years and *n* = 21 participants ≥50 years, *n* = 21) watched a 90 min football match in addition to a 15 min baseline and 15 min halftime break, seated in an environmental chamber (Tair = 31.9 ± 0.4 °C; RH = 76 ± 4%). Gastrointestinal temperature (Tgi), skin temperature (Tskin), and heart rate (HR) were measured continuously throughout the match. Mean arterial pressure (MAP) and perceptual parameters (i.e., thermal sensation and thermal comfort) were scored every 15 min. Tri (37.3 ± 0.4 °C to 37.4 ± 0.3 °C, *p* = 0.11), HR (76 ± 15 bpm to 77 ± 14 bpm, *p* = 0.96) and MAP (97 ± 10 mm Hg to 97 ± 10 mm Hg, *p* = 0.67) did not change throughout the match. In contrast, an increase in Tskin (32.9 ± 0.8 °C to 35.4 ± 0.3 °C, *p* < 0.001) was found. Further, 81% of participants reported thermal discomfort and 87% a (slightly) warm thermal sensation at the end of the match. Moreover, the thermal or cardiovascular responses were not affected by age (all *p*-values > 0.05). Heat stress induced by watching a football match in simulated hot and humid conditions does not result in substantial thermal or cardiovascular strain, whereas a significant perceptual strain was observed.

## 1. Introduction

Major international sporting events (i.e., Paris 2024 Summer Olympics, Australian Tennis Open and US Tennis Open) are often scheduled in thermally challenging environments that may pose a health threat to those participating, officiating, working, and spectating. In such situations, there is a strong focus on ensuring the safety of participating athletes and strategies to optimise their exercise performance [1,2,3,4]. During exercise, only ~20 to 25% of metabolic energy is converted to mechanical work, whereas the remaining energy is released as heat [5]. As a result, metabolic heat production during exercise increases drastically and exceeds the athlete’s heat loss capacity, leading to an increased core temperature and associated performance loss [6]. Therefore, athletes often use heat acclimatisation and/or cooling interventions prior to or during exercise as a strategy to optimise exercise performance in the heat [7,8].

Major sporting events also attract large numbers of spectators, who will also be exposed to thermally challenging environmental conditions. Many of these individuals will be traveling from abroad and are not acclimatised to the local climatic conditions [9,10]. Furthermore, the inherent heterogeneity of visiting spectators (i.e., sex, age, BMI, and medical history) [11,12] and their low accessibility to heat-mitigation measures, such as heat acclimation and cooling strategies, may further enhance the risk of developing heat-related problems and should not be overlooked when organising major competitions [13,14]. In contrast to athletes, spectators are not performing exercise and they are solely exposed to the challenging environmental conditions.

From a physiological perspective, passive heat exposure results in blood flow shifts from central organs (i.e., thoracic and splanchnic region) to the periphery (i.e., skin and large vessels draining the skin) in order to thermoregulate, stimulate heat loss to the environment and maintain core temperature [15]. In the literature, it is demonstrated that passive heat exposure (32–42 °C) elevates heart rate, cardiac output, and skin temperature (T_skin_) [15,16,17,18], whereas core temperature remains unaffected if the environmental conditions are compensable [17]. As airflow in crowded stadiums is limited, and spectators of sporting events are often moderately active (i.e., cheering/stand-up/stadium wave), it is expected that hot environmental conditions may result in increased thermal and cardiovascular strain in unacclimated spectators [19,20]. Moreover, hot ambient temperature reduces the dry heat loss capacity (i.e., convection and radiation), while humid environments reduce the evaporative capacity, which compromises thermal homeostasis and activates the cardiovascular system. Next to the thermal and cardiovascular challenges of passive heat exposure, it might evoke cognitive and psychophysiological challenges as well. It has been demonstrated that an elevation in core temperature can negatively affect cognitive function (i.e., working memory, executive function and attention), while large increments in skin temperature, without associated elevations in core temperature, are related to decreased cognitive function as well [21,22].

That thermally challenging environments can result in heat-related problems for spectators and officials has been illustrated at the 1996 Atlanta Olympics (maximum T_air_ = 35 °C). As much as 13% of the treated spectators and staff at the Olympic venue (*n* = 372) were diagnosed with heat-related illnesses, including heat cramps, dehydration, heat syncope and heatstroke, whereas 10 patients were admitted to the hospital for additional treatment [23]. Other examples of passive heat stress-related heat illness include collapsed ball boys and girls during Wimbledon (2015 and 2019) and the Australian Tennis Open (2014 and 2021). Based on these observations, it might be expected that spectators can experience heat-related problem while spectating. However, more insight into the thermophysiological responses to passive heat exposure while spectating is needed.

Therefore, the aim of our study was to examine the thermal, cardiovascular and perceptual strain of spectators watching a football match in a simulated hot and humid environment. Second, we explored the impact of age on study outcomes because the elderly may have compromised thermoregulation [24,25]. We hypothesised that spectators demonstrate an increase in thermal, cardiovascular and perceptual strain and a disturbed fluid balance. These responses may be exaggerated in older individuals, given the age-related decrements in the capacity to dissipate heat [26]. The outcomes of this study provide valuable information for future event organisations and visiting spectators about the potential risks of heat stress and the associated need for heat-mitigation strategies.

## 2. Materials and Methods

### 2.1. Participants

Participants, aged 16 years or older, were recruited using social media, newspapers, and flyers. Exclusion criteria were based on the use of the ingestible temperature capsule: (I) body weight < 36.5 kg, (II) implanted electro-medical device, (III) history of obstructive/inflammatory bowel disease or surgery, (IV) or a scheduled MRI scan within 5 days of the experiment. Participants were classified into a young (<50 years, *n* = 27) and an older (≥50 years, *n* = 21) group. This group distinction was made based on a previous study demonstrating that the thermoregulatory responses to passive heat exposure are age-dependent [27]. This study was in accordance with the Declaration of Helsinki and was approved by the Medical Ethical Committee of the Radboud university medical center (#2019-5829). All participants gave written informed consent prior to participation.

### 2.2. Study Design

Participants were invited for a single study visit to a climate chamber (B-cat, Tiel, The Netherlands) at the Dutch Olympic Sports Center to watch a match of the Dutch national football team (The Netherlands vs. Spain, FIFA World Cup 2014, Brazil) in hot and humid ambient conditions (T_air_ = 31.9 ± 0.4 °C; RH = 76 ± 4%; water vapor pressure = 3.6 kPa). To mimic the stands in a sports stadium, participants sat in two closely positioned rows of four chairs and watched the football match on a large television screen. Participants were randomly allocated to one of the seats. Participants were asked to wear summer clothing supplemented with typical fan gear (e.g., fan scarf and cap). A total of 6 experimental sessions were scheduled, with 8 participants per session. All experimental sessions were commenced at the same time of day (01:00 p.m. to 05:00 p.m.) to avoid any circadian rhythm effects between sessions.

### 2.3. Study Protocol

Upon arrival, participants gave written informed consent and were instrumented with measurement equipment for the assessment of gastrointestinal temperature (T_gi_), T_skin_, heart rate (HR), and blood pressure. Subsequently, body weight and height measurements were obtained, and participants were asked to provide a midstream urine sample to determine the urine specific gravity (USG). Participants subsequently entered the climate chamber and started with a 15 min seated baseline measurement. Thereafter, participants watched a 90 min football match plus a 15 min halftime break. Participants were allowed to drink ad libitum (i.e., using 0.5 L water bottles and 0.33 L soft drink cans) throughout the experimental trial and stand up and walk within the climate chamber during the halftime break. In addition, blood pressure, thermal sensation, thermal comfort, and state of arousal were measured every 15 min, while T_gi_, T_skin,_ and HR were assessed continuously (Appendix A). Directly after completion of the experimental session, body weight and urine specific gravity were obtained again.

### 2.4. Outcome Parameters

*Thermal strain.* Participants’ T_gi_, a reliable surrogate marker for core temperature, was measured continuously at 30 s intervals using a validated telemetric temperature capsule system (myTemp, Nijmegen, The Netherlands) [28]. Participants ingested the temperature capsule 4 h prior to the experimental session to avoid any interaction with fluid ingestion [29]. Furthermore, T_skin_ was continuously measured at 20 s intervals using wireless temperature recorders (iButton DS1922L, Dallas Semiconductor Corp., Dallas, TX, USA) attached to the skin at 4 distinct locations (i.e., neck, left hand, right shoulder, and right shin) (ISO-9886) using Tegaderm Film (Tegaderm, Neuss, Germany). The resolution was set at 0.0625 °C, and weighted averages were calculated according to international standard operations (ISO-9886) [30].

*Cardiovascular strain.* A V800 HR monitor (Polar Electro Oy, Kempele, Finland) with an H10 sensor chest strap (Polar Electro Oy, Kempele, Finland) were used to measure HR at 15 s intervals throughout the experimental session. Blood pressure was measured once every 15 min using an automated sphygmomanometer (M5-1 IntelliSense, Omron Healthcare, Hoofddorp, The Netherlands).

*Perceptual outcomes.* Participants were instructed to rate their thermal comfort using a 4-point scale ranging from comfortable (1) to very uncomfortable (4) [31]. The thermal sensation was measured using a validated 7-point scale ranging from −3 (cold) to +3 (hot) [31]. State of arousal while watching the football match was scored using an adjusted Visual Analog Scale (VAS), in which the participants were instructed to draw a vertical line on a 10 cm horizontal line [32]. A score of 0 cm represented no excitement at all, and a score of 10 cm represented the highest excitement ever.

*Fluid balance.* Body weight was measured at baseline and after watching the football match using a weighing scale (Seca robusta 813 scale, Hamburg, Germany). Body weight was measured while wearing clothing, fan wear, and measurement equipment. Absolute and relative changes in body weight were calculated after correction for fluid intake, and a relative body weight loss ≥2% was defined as hypohydration [33]. Furthermore, participants were instructed to provide a urine sample at baseline and at the end of the experiment to determine USG using a handheld refractometer (Atago PAL-10S, Atago, Japan) as a measure of hydration status. A USG >1.020 was considered as hypohydration [34].

### 2.5. Statistical Analysis

Minute averages of T_gi_, T_skin,_ and HR were calculated using a customised MATLAB and Statistics Toolbox software package (2012b, The MathWorks, Inc., Natick, MA, USA), and we included a data point every 15 min for our analysis. All statistical analyses were performed using the Statistical Package for Social Sciences software (SPSS v25.0, IBM Corp., Armonk, NY, USA), in which the level of significance was set at *p* < 0.05. All parameters were visually inspected for normality using histogram plots and data are reported as the mean and standard deviation (±SD) or median [interquartile range]. Linear mixed model analysis was used to evaluate the change over time in thermal and cardiovascular parameters, in which age group (young (<50 years) versus old (≥50 years)) was included as a between-subject factor. Differences in baseline characteristics between young and old participants were examined using an unpaired Student’s *t*-test.

## 3. Results

A total of 48 participants (30 males and 18 females, 43 ± 9 years) participated in this study, who were allocated into a young (*n* = 27, 29 ± 11 years) and an older group (*n* = 21, 62 ± 7 years) (Table 1). All participants successfully completed the experimental session. However, T_gi_ measurements were missing in 11 participants due to technical difficulties (not being able to ingest the capsule (*n* = 1), false calibration of temperature capsule (*n* = 1), incomplete data (*n* = 1), and data interference (*n* = 8)), whereas abnormal T_gi_ data from another 3 participants were excluded following the case by case review (Appendix A). Consequently, 34 participants were included in the T_gi_ analysis. Furthermore, heart rate data were missing in two participants due to technical difficulties. Data for all other parameters were available from the entire cohort (*n* = 48).

### 3.1. Thermal Responses

Baseline T_gi_ was 37.3 ± 0.4 °C and did not change over time (*p* = 0.09, Figure 1A), with an end match T_gi_ of 37.4 ± 0.3 °C (Table 2). On the contrary, T_skin_ increased over time (*p* < 0.001, Figure 1B), from 32.9 ± 0.8 °C at baseline to 35.4 ± 0.3 °C at the end of the match. The young group demonstrated a higher T_gi_ (37.4 ± 0.3 °C vs. 37.1 ± 0.3 °C, *p* < 0.001) and T_skin_ (35.3 ± 0.3 °C vs. 35.0 ± 0.4 °C, *p* < 0.001) throughout the match compared to the older group, whereas comparable responses in T_gi_ and T_skin_ were found across groups over time (*p* = 0.38 and *p* = 0.62, respectively, Figure 1). Moreover, the increase in T_skin_ throughout did not differ between males and females (*p* = 0.80, Appendix A).

### 3.2. Cardiovascular Responses

Baseline HR was 75 ± 15 bpm and did not change throughout the match (*p* = 0.97, Figure 2A, Table 2). Baseline systolic blood pressure, diastolic blood pressure, and MAP were 126 ± 13 mm Hg, 83 ± 9 mm Hg, and 97 ± 10 mm Hg, respectively, and did not change over time (*p* = 0.65, *p* = 0.84, and *p* = 0.67, respectively, Figure 2B). The young group demonstrated a higher HR (81 ± 12 bpm vs. 68 ± 11 bpm, *p* < 0.001), a lower systolic blood pressure (121 ± 13 mm Hg vs. 127 ± 12 mm Hg, *p* < 0.001), a lower diastolic blood pressure (78 ± 10 mm Hg vs. 85 ± 9 mm Hg, *p* < 0.001), and a lower MAP (92 ± 10 mm Hg vs. 99 ± 9 mm Hg, *p* < 0.001) throughout the match compared to the older group. No sex differences in HR and MAP response throughout the match were found (*p* = 0.28 and *p* = 0.58, respectively, Appendix A).

### 3.3. Perceptual Outcomes

Thermal comfort and thermal sensations at baseline were 2 [1–3] and 1 [−1–3], respectively. At baseline, 52% of younger and 38% of older participants reported thermal discomfort, which increased to 56% and 71% at the end of the match, respectively (Figure 3). A (slightly) warm thermal sensation was reported by 89% of younger and 91% older participants at baseline and by 85% of the younger and 91% of the older participants at the end of the match (Figure 3). The baseline state of arousal was 1.2 ± 1.6 and increased throughout the match to a peak value of 4.4 ± 2.5 au (*p* < 0.001).

### 3.4. Fluid Balance

Fluid intake was 0.8 ± 0.2 L and did not differ between age groups (*p* = 0.71). Based on body mass loss, participants were, as a whole, hypohydrated by 0.1 ± 0.2 kg, but, according to USG, could be classified as euhydrated. Baseline USG was 1.013 ± 0.007 g/mL and decreased during the match to 1.008 ± 0.006 g/mL (*p* < 0.001). Body weight loss (*p* = 0.21) and baseline USG did not differ between age groups (*p* = 0.40), but post-match urine specific gravity was lower in the young (1.007 ± 0.005 g/mL) compared to the old group (1.010 ± 0.007 g/mL, *p* = 0.033) and 4% (*n* = 1) of the young group and 14% (*n* = 3) of the old group were considered dehydrated (*p* = 0.19).

## 4. Discussion

This study examined the thermal, cardiovascular, and perceptual responses encountered by spectators watching a football match in a simulated hot and humid environment. In contrast to our hypothesis, no significant changes in T_gi_, HR, and MAP were observed while watching a football match in the heat, whereas a progressive rise in T_skin_ and significant perceptual strain were observed. The responses were not affected by age. These findings indicate that watching a football match in thermally challenging environmental conditions (T_air_ = 31.9 ± 0.4 °C; RH = 76 ± 4%) does not result in substantial thermal or cardiovascular strain, whereas significant perceptual strain was observed.

### 4.1. Thermal Strain

T_gi_ increased with 0.1 °C while watching a football match in the heat in the present study. This observation is contradictory to a previous lab-based study which reported a ~0.6 °C increase in sublingual temperature (a surrogate marker for core temperature) following 90 min of passive heat exposure (40 °C, 43% RH) [27]. Differences in ambient temperature (32 °C vs. 40 °C) and/or measurement technique (gastrointestinal vs. sublingual) may explain these discrepant outcomes, suggesting that the simulated hot and humid conditions in the present study reflect compensable heat strain. Indeed, ambient temperature was constantly lower than T_skin_, allowing dry heat loss to the environment [35,36]. The observed increase in T_skin_ further reinforces this hypothesis, as vasodilation of the skin arterioles enhances passive heat loss [16,35,37]. This dry heat loss pathway is particularly important in thermally challenging hot and humid conditions as the evaporative heat loss capacity is diminished due to high absolute humidity [38,39]. Taken together, our data demonstrate that spectators can preserve thermal homeostasis in a simulated hot and humid environment (T_air_ = 31.9 ± 0.4 °C; RH = 76 ± 4%).

### 4.2. Cardiovascular Strain

In the present study, HR and MAP did not change, suggesting that spectators did not encounter significant cardiovascular strain. Previous studies found that cardiac output may increase 2.5-fold during passive heat exposure to allow maximal skin perfusion, which can be accomplished by increases in HR [16,37]. The magnitude of change in cardiac output and heart rate is dependent on the severity and duration of the heat exposure [16], potentially indicating that ambient conditions in our study were not severe enough to induce cardiovascular strain. The state of arousal may impact cardiovascular strain as well [40], but excitement levels were relatively low in our study because participants watched a historic football match instead of being part of a live sporting event in a sports stadium. Therefore, cardiovascular strain is presumably underestimated in the present study.

### 4.3. Perceptual Strain

The link between perceptual outcomes and thermal strain is well established. At a normothermic core temperature (~37 °C), thermal sensation and thermal comfort are largely determined by T_skin_ [41,42,43], whereas thermal comfort might also be modulated by skin wettedness [43,44,45]. We found a progressive increase in T_skin_, with peak values of 35.4 ± 0.3 °C at the end of the match. This increase in T_skin_ may explain the parallel observation of significant perceptual strain, as 63% of participants reported thermal discomfort and 88% a (slightly) warm thermal sensation at the end of the match.

### 4.4. Fluid Balance Responses

Average body weight loss was 0.1 ± 0.2 kg, and none of the participants were hypohydrated after watching the football match. Participants were allowed to drink ad libitum during the experiment, and the average fluid intake was 0.8 ± 0.2 L. Therefore, our results indicate that spectators are well able to preserve fluid balance in the heat when fluid intake is unlimited available. Importantly, the risk of hypohydration may increase when exposure times are prolonged (prior to, during, and after the sporting event) and alcohol is consumed, highlighting the importance of staying hydrated throughout the day.

### 4.5. Young versus Older Adults

It has been shown that cutaneous vasodilation, thermal sensitivity, sweat rate, and the redistribution of blood from internal organs are blunted in the elderly during passive heat exposure [46,47]. Given these age-related decrements in the capacity to dissipate heat [26], we expected to see exaggerated thermoregulatory and cardiovascular responses (i.e., T_gi_, T_skin_, HR, and MAP) in the older group. However, although we did observe a significantly higher T_skin_ in younger individuals, the observed thermoregulatory and cardiovascular responses in our study were not affected by age. The absence of any age-related impact on our study outcomes is most likely because the overall thermoregulatory and cardiovascular strain was already limited. Further, older individuals are more likely to have health issues (e.g., cardiovascular and renal diseases), which further compromise thermoregulation [48], but our study consisted of a relatively healthy group of older individuals.

### 4.6. Strengths and Limitations

To the best of our knowledge, our study is the first to examine the effects of watching a football match in simulated hot and humid environmental conditions on thermoregulatory, cardiovascular, perceptual, and fluid balance responses. However, some limitations should be considered. First, the experimental sessions were performed in a climate chamber. Participants were not exposed to the sun, limiting radiant heat and associated heat transfer, which may have reduced thermal strain and underestimated the physiological responses to hot and humid conditions outdoors. On the other hand, airflow in the present study was lower than expected in outdoor venues, which may (partially) neutralise the lack of sun exposure. Second, participants were exposed to a hot and humid climate for only 2 h, whereas the exposure time during real-life major sporting events is expected to be much longer (i.e., travel time, multiple games a day) for multiple days. Further, during real-life sporting events, spectators may be exposed to additional personal and environmental factors (i.e., alcohol intake, hypohydration, poor nutrition, lack of sleep, and sunburns) that increase the risk of developing heat-related illnesses. Together with the environmental heat stress, these factors could exacerbate the observed thermal, cardiovascular, perceptual and fluid balance responses. Third, we did not collect data on the phase of the menstrual cycle of the female participants. A previous study demonstrated that during passive heat exposure there are no sex differences in metabolic heat production, local sweat rate, number of activated sweat glands and cutaneous blood flow, while the menstrual cycle modified the core temperature threshold for vasodilation and the onset of sweating [49]. Therefore, future studies are needed to examine whether the phase of the menstrual cycle impacts the thermophysiological response to passive heat exposure. Fourth, no information was available about the habitual physical activity levels of the participants. Although a higher aerobic fitness is associated with a higher heat loss capacity [50], the passive nature of our study did not require maximal heat loss capacity nor sweat rates of participants. Given the low magnitude of thermal stress in our study, it can be assumed that the thermoregulatory benefits of a higher aerobic fitness would be limited at best.

In future studies, the results of this laboratory study should be confirmed in field studies during actual sporting events, which makes it possible to account for sun exposure, airflow, longer exposures, and additional personal and environmental factors (i.e., alcohol intake, hypohydration, poor nutrition, lack of sleep, and sunburns). Moreover, future studies should obtain the habitual physical activity levels of the participants in order to examine the impact of activity level on thermoregulatory, cardiovascular and perceptual responses to passive heat exposure.

## 5. Conclusions

Heat stress induced by watching a football match in simulated hot and humid conditions (T_air_ = 31.9 ± 0.4 °C; RH = 76 ± 4%) does not result in substantial thermal or cardiovascular strain, whereas a significant perceptual strain was observed. The thermal or cardiovascular responses were not affected by age. Findings from this study indicate that watching a football match for 120 min in a thermally challenging environmental condition may be uncomfortable but is generally safe for unacclimatised individuals.

## Figures and Tables

**Figure 1 sports-11-00078-f001:**
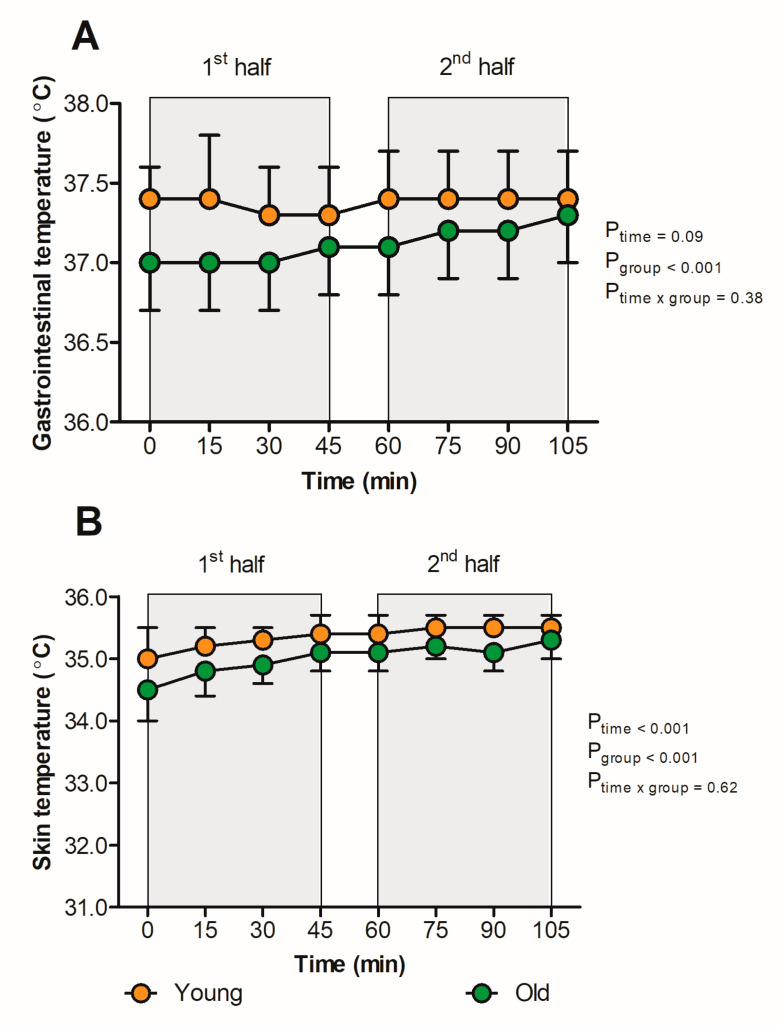
Responses of T_gi_ (**A**) and T_skin_ (**B**) in young (<50 years; orange icons) versus old (≥50 years, green icons) individuals while spectating a football match in hot and humid conditions. In short, T_gi_ and T_skin_ were significantly higher in the young versus the old group, whereas the change in T_gi_ and T_skin_ over time did not differ between groups. Data were presented as the mean ± SD.

**Figure 2 sports-11-00078-f002:**
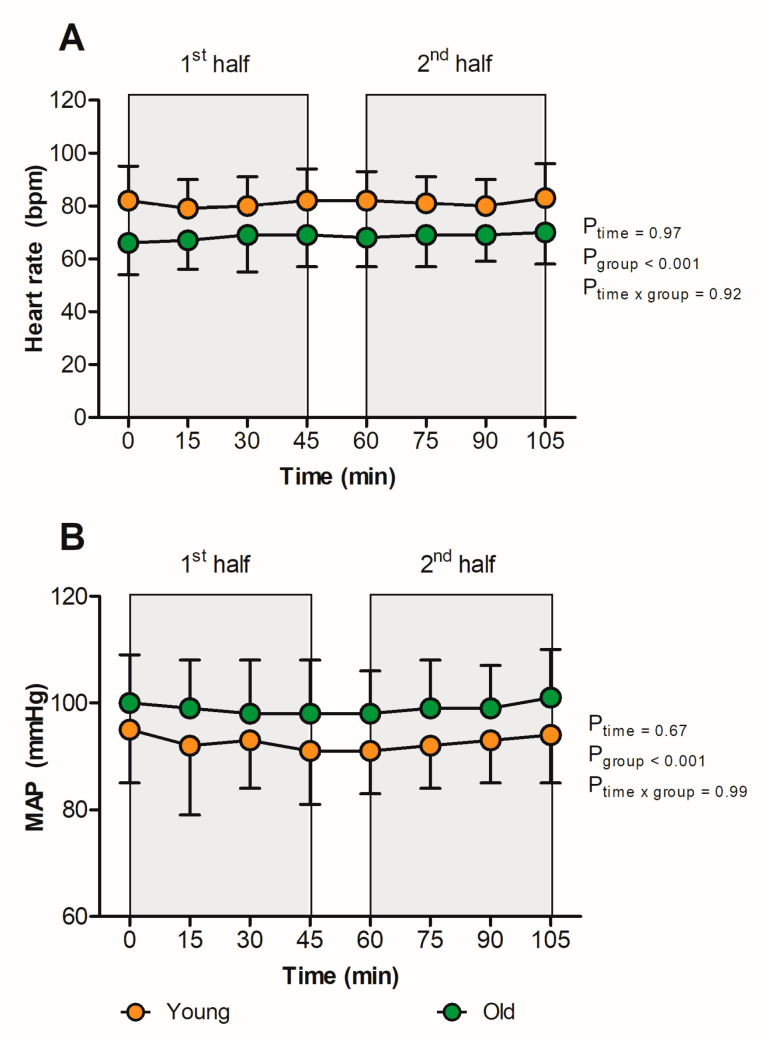
Responses of HR (**A**) and MAP (**B**) in the young (<50 years; orange icons) versus the old group (≥50 years, green icons) while spectating a football match in hot and humid conditions. In short, HR and MAP were significantly lower in the young versus the old group, whereas the change in HR and MAP over time did not differ between groups. Data were presented as the mean ± SD.

**Figure 3 sports-11-00078-f003:**
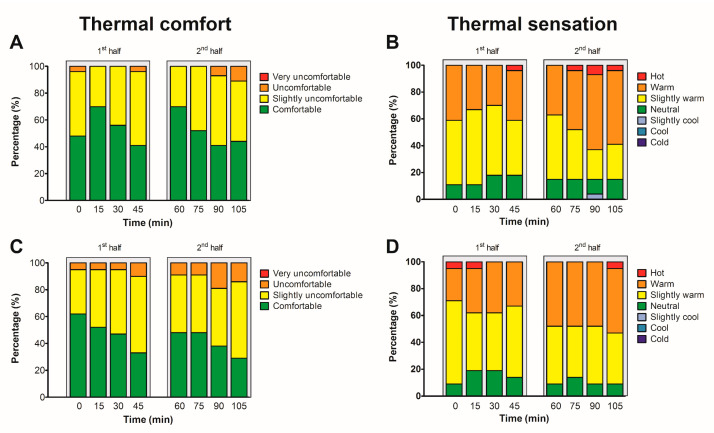
Thermal comfort (panels **A** + **C**) and thermal sensation (panel **B** + **D**) for the young (<50 years, **A** + **B**) and old groups (≥50 years, **C** + **D**) while spectating a football match in simulated hot and humid conditions.

**Table 1 sports-11-00078-t001:** Participant characteristics.

	Total Group (*n* = 48)	Young Group (*n* = 27)	Old Group (*n* = 21)	*p*-Value
Age (years)	43 ± 19	29 ± 11	62 ± 7	<0.001
Sex (*n*(%) male)	30 (63%)	14 (52%)	16 (76%)	0.08
Weight (kg)	77.8 ± 12.4	75.2 ± 11.8	81.3 ± 12.7	0.09
Height (cm)	181 ± 9	181 ± 10	180 ± 8	0.66
BMI (kg/m^2^)	23.8 ± 2.8	22.8 ± 2.4	25.0 ± 2.9	0.006

**Table 2 sports-11-00078-t002:** Thermal and cardiovascular outcome parameters.

	Total Group (*n* = 48)	Young Group (*n* = 27)	Old Group (*n* = 21)	*p*-Value
Baseline T_core_ (°C)	37.3 ± 0.4	37.5 ± 0.3	37.0 ± 0.3	<0.001
Peak T_core_ (°C)	37.5 ± 0.3	37.6 ± 0.3	37.3 ± 0.3	0.005
Baseline T_skin_ (°C)	32.8 ± 0.8	32.9 ± 0.8	32.7 ± 0.8	0.66
Peak T_skin_ (°C)	35.4 ± 0.3	35.6 ± 0.2	35.3 ± 0.2	0.001
Baseline HR (bpm)	76 ± 15	82 ± 14	68 ± 13	0.007
Peak HR (bpm)	81 ± 14	86 ± 12	75 ± 14	0.022
Baseline MAP (mmHg)	96 ± 8	92 ± 5	101 ± 9	0.003
Peak MAP (mmHg)	100 ± 8	97 ± 6	104 ± 9	0.015

## Data Availability

Data are available upon reasonable request.

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
