# Peer review of "Thermoregulatory, Cardiovascular and Perceptual Responses of Spectators of a Simulated Football Match in Hot and Humid Environmental Conditions"

_sports, 2023, doi:10.3390/sports11040078_

Round 1
Reviewer 1 Report (New Reviewer)
In general, it can be concluded that this work is valuable and that you can examine the spectators in a competitive environment as well, which is one of the influential components in a competitive environment, is of utmost importance. However, there are some things that it seems, following them can help to further strengthen your work, especially some variables affecting the research results should be examined in the section of introduction and discussion, which is mentioned below.
It is true that you have investigated the physiological responses of the spectators, but I recommend that you also mention the physiological responses of the players in the introduction section to some extent, that finally reporting the results of your research on spectators can create a comparable criterion implicitly. In this regard, you can use the following sources
Halouani, Jamel, et al. "Technical analysis and heart rate response of minifootball players during a competitive match." International Journal of Sport Studies for Health 3.2 (2020).
Your final result is as follows:
"Heat stress induced by watching a football match in simulated hot and humid conditions does not result in substantial thermal or cardiovascular strain, whereas a significant perceptual strain was observed."
As it was clear from the beginning, there is a direct relationship between physiological responses and cognitive and psychomotor performance, and you have correctly used the word perception, but you did not mention the relationship between physiological and cognitive capacities in the introduction. You must address this issue and you can. Use the following references:
Soylu, Yusuf, Ersan Arslan, and Bulent Kilit. "Psychophysiological responses and cognitive performance: a systematic review of mental fatigue on soccer performance." International Journal of Sport Studies for Health 4.2 (2021).
Halouani, Jamel, et al. "Physical and Heart Rate Responses of Male Minifootball Players: A Case Study of an Elite Minifootball Match." International Journal of Sport Studies for Health 3.2 (2020).
Adjust the keywords according to the open mesh standard
P values should be accurately reported in the results section of the abstract and try not to include keywords in the research title.
While the variables in your research results are as follows:
"thermoregulatory, cardiovascular, perceptual, and fluid balance responses"
you have only mentioned the physiological issue in the title, so it seems better to change the title based on the mentioned variables.
There is a clear difference between a real football match and a simulated match, which should be discussed in the discussion. You can use the following sources.
Hsouna, Hsen, et al. "A daytime 40-min nap opportunity after a simulated late evening soccer match reduces the perception of fatigue and improves 5-m shuttle run performance." Research in Sports Medicine 30.5 (2022): 502-515.
It is suggested that the work execution steps be made in the form of a flowchart
It seems that there are some works of ratio similar to this work, which can be the basis of comparison in the discussion and conclusion
What is the suggestion for future researchers? Make suggestions based on the limitations of the research
Author Response
See attachement for rebuttal and figure
Responses to reviewer 1:
In general, it can be concluded that this work is valuable and that you can examine the spectators in a competitive environment as well, which is one of the influential components in a competitive environment, is of utmost importance. However, there are some things that it seems, following them can help to further strengthen your work, especially some variables affecting the research results should be examined in the section of introduction and discussion, which is mentioned below.
We would like to thank the reviewer for his/her constructive feedback on our manuscript and we have addressed the raised issues below.
- It is true that you have investigated the physiological responses of the spectators, but I recommend that you also mention the physiological responses of the players in the introduction section to some extent, that finally reporting the results of your research on spectators can create a comparable criterion implicitly. In this regard, you can use the following sources:
- Halouani, Jamel, et al. "Technical analysis and heart rate response of minifootball players during a competitive match." International Journal of Sport Studies for Health 3.2 (2020).
We agree with the reviewer that it is valuable to mention the physiological responses of athletes to exercise in the heat. We have included the following sentences in the introduction (Line 36-42):
“During exercise, only ~20 to 25% of metabolic energy is converted to mechanical work, whereas the remaining energy is released as heat [5]. As a result, the metabolic heat production during exercise increases drastically and exceeds the athlete’s heat loss capacity, leading to an increased core temperature and associated performance loss [6]. Therefore, athletes often use heat acclimatisation and/or cooling interventions prior to or during exercise as a strategy to optimize exercise performance in the heat [7, 8]”.
- Your final result is as follows: "Heat stress induced by watching a football match in simulated hot and humid conditions does not result in substantial thermal or cardiovascular strain, whereas a significant perceptual strain was observed."
As it was clear from the beginning, there is a direct relationship between physiological responses and cognitive and psychomotor performance, and you have correctly used the word perception, but you did not mention the relationship between physiological and cognitive capacities in the introduction. You must address this issue and you can. Use the following references:
- Soylu, Yusuf, Ersan Arslan, and Bulent Kilit. "Psychophysiological responses and cognitive performance: a systematic review of mental fatigue on soccer performance." International Journal of Sport Studies for Health 4.2 (2021).
- Halouani, Jamel, et al. "Physical and Heart Rate Responses of Male Minifootball Players: A Case Study of an Elite Minifootball Match." International Journal of Sport Studies for Health 3.2 (2020).
The reviewer raises a fair point. We have now addressed the cognitive and psychophysiological performance in the introduction. It reads (Line 64-69):
“Next to the thermal and cardiovascular challenges of passive heat exposure, it might evoke cognitive and psychophysiological challenges as well. It has been demonstrated that an elevation in core temperature can negatively affect cognitive function (i.e., working memory, executive function and attention), while large increments in skin temperature, without associated elevations in core temperature, is related with a de-creased cognitive function as well [21, 22].”
However, based on the scope of our manuscript (spectators of live sporting events) we described the effect of passive heat exposure on psychophysiological and cognitive performance, and we did not include the psychophysiological responses and cognitive performance of participating athletes.
- Adjust the keywords according to the open mesh standard
We have now adjusted the keywords to the open mesh standards: Body temperature regulation, sports, extreme heat, heat stress disorders, hydration status.
- P values should be accurately reported in the results section of the abstract.
We have checked our abstract and added missing p-values in the results section of the abstract. It now reads (Line 20-25):
“Tgi (37.3±0.4°C to 37.4±0.3°C, p=0.11), HR (76±15 bpm to 77±14 bpm, p=0.96) and MAP (97±10 mm Hg to 97±10 mm Hg, p=0.67) did not change throughout the match. In contrast, an increase in Tskin (32.9±0.8°C to 35.4±0.3°C, p<0.001) was found. Besides, 81% of participants reported thermal discomfort and 87% a (slightly) warm thermal sensation at the end of the match. Moreover, the thermal or cardiovascular responses were not affected by age (all p-values >0.05).”
- Try not to include keywords in the research title.
We have checked the title of our manuscript and the keywords used, and we did not find any similarities in both.
- While the variables in your research results are as follows: "thermoregulatory, cardiovascular, perceptual, and fluid balance responses" you have only mentioned the physiological issue in the title, so it seems better to change the title based on the mentioned variables.
We have changed the title of our manuscript. It now reads: “Thermoregulatory, cardiovascular and perceptual responses of spectators of a simulated football match in hot and humid environmental conditions”.
- There is a clear difference between a real football match and a simulated match, which should be discussed in the discussion. You can use the following sources:
- Hsouna, Hsen, et al. "A daytime 40-min nap opportunity after a simulated late evening soccer match reduces the perception of fatigue and improves 5-m shuttle run performance." Research in Sports Medicine 30.5 (2022): 502-515.
We agree with the reviewer that there is a difference between a real-life sporting event and a simulated match. We have addressed this as follows (Line 298-304):
“Second, participants were exposed to a hot and humid climate for only 2 hours, whereas the exposure time during real-life major sporting events is expected to be much longer (i.e., travel time, multiple games a day) for multiple days. Besides, during real-life sporting events, spectators may be exposed to additional personal and environmental factors (i.e., alcohol intake, hypohydration, poor nutrition, lack of sleep, and sunburns) that increase the risk of developing heat-related illnesses.”
After thoroughly studying the paper that has been suggested by the reviewer, we do not believe that it can be used as a reference in our manuscript. In the study by Hsouna et al. the researchers examined the impact of a daytime 40-min nap following an evening simulated soccer match on exercise performance, muscle soreness, perception of sleepiness and rate of perceived exertion. Thus, the simulated soccer match was used as an intervention to examine the primary outcome variables the next day. While we examined our primary outcome measures (i.e., thermoregulatory, cardiovascular and perceptual responses) during the simulated match.
Therefore, we decided to not include this reference in our manuscript.
- It is suggested that the work execution steps be made in the form of a flowchart
We want to thank the reviewer for this suggestion and have now added a flowchart of the experimental design as Supplemental Figure 1.
- It seems that there are some works of ratio similar to this work, which can be the basis of comparison in the discussion and conclusion
The reviewer is right that there are studies that examine thermoregulatory, cardiovascular and perceptual responses to passive heat exposure and we have used the following studies in our discussion section:
- Crandall, C.G., et al., Effects of passive heating on central blood volume and ventricular dimensions in humans. J Physiol, 2008. 586(1): p. 293-301.
- Crandall, C.G. and T.E. Wilson, Human cardiovascular responses to passive heat stress. Compr Physiol, 2015. 5(1): p. 17-43.
- Dufour, A. and V. Candas, Ageing and thermal responses during passive heat exposure: sweating and sensory aspects. European Journal of Applied Physiology, 2007. 100(1): p. 19-26.
- Vargas, N.T., et al., The motivation to behaviorally thermoregulate during passive heat exposure in humans is dependent on the magnitude of increases in skin temperature. Physiol Behav, 2018. 194: p. 545-551.
- What is the suggestion for future researchers? Make suggestions based on the limitations of the research
Thank you for this valuable suggestion. We have added the following suggestions for future studies to the discussion of our manuscript (Line 310-312 and Line 318-324):
“Therefore, future studies are needed to examine whether the phase of the menstrual cycle impacts the thermophysiological response to passive heat exposure.”
“In future studies the results of this laboratory study should be confirmed in field studies during actual sporting events, which makes it possible to account for sun exposure, airflow, longer exposures, and additional personal and environmental factors (i.e., alcohol intake, hypohydration, poor nutrition, lack of sleep, and sunburns). Moreover, future studies should obtain the habitual physical activity levels of the participants in order to examine the impact of activity level on thermoregulatory, cardiovascular and perceptual responses to passive heat exposure.”

Reviewer 2 Report (Previous Reviewer 2)
please find attached

Author Response
Response to Reviewer 2
I am happy with the changes. I thought from the beginning that this was a good study to publish.
We would like to thank the reviewer for his/her constructive feedback on our manuscript and we have addressed the raised issues below.
- The way you are presenting your limitations is indicating that you don’t think they are limitations, while in reality, they are. For example, Line 300: I don’t think there is a need to indicate studies of the differences between males and females regarding thermoregulation. I believe your statement is inaccurate. Indeed, most studies examine thermoregulation during exercise, but that doesn’t mean that men and women respond the same during passive heat exposure. Even in the study, you are referring to (47) there is evidence of an effect of the menstrual cycle on thermoregulation. The authors indicated: The menstrual cycle modified the T(b) threshold for vasodilation and sweat onset in women. Therefore, the sex difference in the T(b) threshold was more marked for women during the L phase than during the F phase. Moreover, the menstrual cycle modified the slope of the relationship between %LDF on the back and T(b).
The reviewer makes a fair point regarding the impact of the menstrual cycle on the thermoregulatory and cardiovascular responses to passive heat exposure. We have now rephrased this part of the discussion. (Line 305-312).
“Third, we did not collect data on the phase of the menstrual cycle of the female participants. A previous study demonstrated that during passive heat exposure there are no sex differences in metabolic heat production, local sweat rate, number of activated sweat glands and cutaneous blood flow, while the menstrual cycle modified the core temperature threshold for vasodilation and the onset of sweating [49]. Therefore, future studies are needed to examine whether the phase of the menstrual cycle impacts the thermophysiological response to passive heat exposure.”
- Lastly, since you are including males and females in the study it is expected to present the results of males and females separately in a table. Even if they are not significantly different.
Following the suggestion of the reviewer, we have added a table with sex specific outcomes to the manuscript (see below) and included the following statement in the results section (Line 190-191 and Line 201-203):
“Moreover, the increase in Tskin throughout did not differ between males and females (p=0.80, Supplementary Table 1).”
“No sex differences in HR and MAP response throughout the match were found (p=0.28 and p=0.58, respectively, Supplementary Table 1).”
|
Supplementary Table 1. Sex differences in thermal and cardiovascular outcome parameters |
|||||
|
|
Total group |
Males |
Females |
P-value |
|
|
Age (years) |
43 ± 19 |
47 ± 18 |
36 ± 18 |
0.04 |
|
|
Baseline Tcore (°C) |
37.3 ± 0.4 |
37.1 ± 0.7 |
37.5 ± 0.3 |
<0.004 |
|
|
Peak Tcore (°C) |
37.5 ± 0.3 |
37.4 ± 0.3 |
37.6 ± 0.3 |
0.10 |
|
|
ΔTcore (°C) |
0.2 ± 0.2 |
0.3 ± 0.0 |
0.1 ± 0.1 |
0.001 |
|
|
Baseline Tskin (°C) |
32.8 ± 0.8 |
32.9 ± 0.9 |
32.9 ± 0.6 |
0.92 |
|
|
Peak Tskin (°C) |
35.4 ± 0.3 |
35.4 ± 0.3 |
35.5 ± 0.3 |
0.67 |
|
|
ΔTskin (°C) |
2.6 ± 0.8 |
2.5 ± 0.9 |
2.6 ± 0.6 |
0.80 |
|
|
Baseline HR (bpm) |
76 ± 15 |
72 ± 16 |
81 ± 11 |
0.041 |
|
|
Peak HR (bpm) |
81 ± 14 |
78 ± 14 |
87 ± 12 |
0.05 |
|
|
ΔHR (bpm) |
6 ± 7 |
6 ± 8 |
4 ± 5 |
0.28 |
|
|
Baseline MAP (mmHg) |
96 ± 8 |
100 ± 10 |
93 ± 9 |
0.017 |
|
|
Peak MAP (mmHg) |
100 ± 8 |
103 ± 9 |
97 ± 10 |
0.033 |
|
|
ΔMAP (mmHg) |
4 ± 5 |
3 ± 4 |
4 ± 5 |
0.58 |
|

Round 2
Reviewer 1 Report (New Reviewer)
ACCEPTED.
This manuscript is a resubmission of an earlier submission. The following is a list of the peer review reports and author responses from that submission.
Round 1
Reviewer 1 Report
Although the article is related to sports, I think it does not fall within the scope of the journal. On the other hand, there are some flaws in the article. In addition, psychological variables are involved when observing a simulated match, which are not the same as in a match in a real situation, where the spectator can move more due to the animation of the team or the celebration of a goal.
Introduction. It is very short and should justify the realization of why the research was carried out. It should also provide information on the thermoregulatory processes that take place after exposure.
Methods. In this study 18 women participate, where the menstrual phase they are in is not specified. It is known that the phases of the cycle affect the variables that the authors measure, such as temperature. Therefore, it is important to know the menstrual phase in which they are. Another important aspect that affects the variables is the physical activity carried out by the participants, so it would have been necessary to know this data by means of a physical test or the IPAQ questionnaire. On the other hand, the statistical test used is very basic; an ANOVA could have been performed.
Results. The results should be presented in tables knowing the p-value, the most important results being those that stand out graphically.
Author Response
The response to the reviewers suggestions are given in the attached file.

Reviewer 2 Report
First, I would like to recognize the authors for the data they collected to examine the thermal, cardiovascular, and perceptual responses of spectators watching a football match in a simulated hot and humid environment. The results of this well-controlled study are interesting as major sporting events are often organized in hot and humid environments. Most studies pay attention to the players/athletes only but not the people working at those events or spectating.
Abstract
The abstract provides the reader with the purpose, methods, results, and conclusions.
Introduction
The introduction is well developed leading to the significance, the need, and the purpose of the study. The introduction is short and right to the point without unnecessary information.
Methods
The materials and methods section is presented with sufficient detail so that someone can replicate and build on the published results. In addition, the methods were appropriately presented and cited. I really like and support the methodology by which the perceptual outcomes were collected as perceptual responses tend to be problematic in several studies.
Results
At one point it was confusing as you have presented the males and females in the abstract and then in the methodology you have presented young and old participants. It wasn’t clear as to how many young and old males or females were participating but that was cleared on Table 1.
Discussion
The results were discussed, interpreted, and compared to previous studies in the discussion section. Finally, the limitations are clearly stated. The only comment I have for the study is the lack of comparison between males and females. In the abstract, you presented males and females and gave the impression that a comparison was going to follow between genders. However, the only comparison that was presented was between young and old individuals. This is a minor issue but I would either add a short comparison between males and females or in the abstract report the young and old participants and present the number of males and females in the methodology or results sections only.
Author Response

(The authors gave the same response as above.)
